# The Time Trajectory of Choroid Plexus Enlargement in Multiple Sclerosis

**DOI:** 10.3390/healthcare12070768

**Published:** 2024-04-01

**Authors:** Athina Andravizou, Sotiria Stavropoulou De Lorenzo, Evangelia Kesidou, Iliana Michailidou, Dimitrios Parissis, Marina-Kleopatra Boziki, Polyxeni Stamati, Christos Bakirtzis, Nikolaos Grigoriadis

**Affiliations:** 1Multiple Sclerosis Center, Second Department of Neurology, School of Medicine, Aristotle University of Thessaloniki, 54621 Thessaloniki, Greece; athina.andravizou9@hotmail.com (A.A.); iradel7714@gmail.com (S.S.D.L.); kesidoue@auth.gr (E.K.); ilianam@auth.gr (I.M.); dparisis@auth.gr (D.P.); bozikim@auth.gr (M.-K.B.); ngrigoriadis@auth.gr (N.G.); 2Department of Neurology, University Hospital of Larissa, 41334 Larissa, Greece; tzeni_0@yahoo.gr

**Keywords:** choroid plexus, multiple sclerosis, inflammation, neurodegeneration, volumetric, EDSS

## Abstract

Choroid plexus (CP) can be seen as a watchtower of the central nervous system (CNS) that actively regulates CNS homeostasis. A growing body of literature suggests that CP alterations are involved in the pathogenesis of multiple sclerosis (MS) but the underlying mechanisms remain elusive. CPs are enlarged and inflamed in relapsing-remitting (RRMS) but also in clinically isolated syndrome (CIS) and radiologically isolated syndrome (RIS) stages, far beyond MS diagnosis. Increases in the choroid plexus/total intracranial volume (CP/TIV) ratio have been robustly associated with increased lesion load, higher translocator protein (TSPO) uptake in normal-appearing white matter (NAWM) and thalami, as well as with higher annual relapse rate and disability progression in highly active RRMS individuals, but not in progressive MS. The CP/TIV ratio has only slightly been correlated with magnetic resonance imaging (MRI) findings (cortical or whole brain atrophy) and clinical outcomes (EDSS score) in progressive MS. Therefore, we suggest that plexus volumetric assessments should be mainly applied to the early disease stages of MS, whereas it should be taken into consideration with caution in progressive MS. In this review, we attempt to clarify the pathological significance of the temporal CP volume (CPV) changes in MS and highlight the pitfalls and limitations of CP volumetric analysis.

## 1. Introduction

Central nervous system (CNS) homeostasis is established by endothelial, epithelial, meningeal, and glial cell barriers that divide the brain into compartments [1]. Choroid plexus (CP) is located within the four ventricles of the brain, at the intersection of systemic blood circulation and the cerebrospinal fluid (CSF) compartment [2]. Structurally, the plexus is a delicate, continuous layer of epithelial cells connected by tight junctions and networks of capillary vessels separated from the epithelial cells by a connective tissue stroma [3]. Choroidal capillaries are less fenestrated compared to cerebral vessels, allowing the diffusion of certain molecules into the plexus epithelium. Both the blood–CSF barrier (BCSFB) (the choroid plexus epithelium) and the blood–brain barrier (BBB) (choroidal capillary vessels) regulate the influx of cells and solutes from the blood to the CSF and into the brain parenchyma [4]. 

The plexus is important not only for the production of CSF but also for brain waste removal through the “glymphatic clearance” (a recently described glial cell supported-lymphatic system) of CSF along the periarterial spaces [5,6]. Furthermore, the CP is also responsible for regulating the entry of immune cells into the brain parenchyma [7,8]. In the healthy brain, several kinds of immune cells reside in CPs, mainly located in the plexus stroma and epithelium, and provide the first line of deference against CNS inflammation [9,10]. These include myeloid cells and mostly macrophages (Kolmer cells), while the lymphocyte population in the plexus remains unclear [4,11]. In the healthy brain, B-lymphocytes and plasma cells seem to be absent from the CP [11].

Additional functions of the CP include the active protection of the brain and spinal cord via the secretion of neurotrophic and angiogenic factors from epithelial cells into the CSF. These factors include insulin-like growth factor 2 (IGF2), transforming growth factor-b (TGF-b), and transthyretin to name but a few [6,12]. Moreover, the plexus is involved in neuronal repair processes, as well as in the restoration of the brain environment after trauma or stroke via the expression of epithelial growth factors and the modulation of immune cell content [6]. Figure 1 depicts the function and architecture of CPs. 

While our understanding of the active role of CP in brain homeostasis and immunosurveillance is still growing, much less is known about its role in MS pathophysiology. Neuroinflammation is observed in all forms of MS; however, the transient infiltration of peripheral lymphocytes through a dysfunctional BBB into the CNS is considered critical particularly during the early disease stages, such as clinically isolated syndrome (CIS) or relapsing–remitting (RR) MS. As the disease progresses, the neuroinflammatory properties change, with later stages characterized by mild inflammation, commonly referred as “slow-burning” or intrathecal inflammation, behind a relatively intact or restored BBB [13]. Both forms of progression, either primary (PP) or secondary progressive (SP) MS, together accounting for around 20% of MS individuals, essentially reflect the neurodegenerative component [4,13]. In this review, we attempt to explore the pathological significance of CP enlargement in MS since it has been associated with both neuroinflammation and neurodegeneration across the MS spectrum.

## 2. Materials and Methods

We reviewed the most recent evidence regarding the CP enlargement in the immune pathophysiology of MS. Articles for this review were identified through the Medline, Google Scholar, and Scopus databases with the search terms “choroid plexus”, “multiple sclerosis”, “inflammation”, “neurodegeneration”, “volumetric” and “EDSS”. All authors participated in the search for the available literature. Posters presented at main congresses in the field of MS were also evaluated. We included only papers written in English. Original studies were thoroughly reviewed with regard to the methodology used, for the volumetric analysis of the choroid plexi. The final reference list was generated on the basis of originality, and relevance to the broad spectrum of this review, with a particular focus on papers published in the last 5 years. 

## 3. CP Enlargement as a Biomarker of Disease Activity

Several recent studies have drawn attention to the unanticipated involvement of CP in individuals with MS [14,15,16,17,18,19,20]. The study of human CP morphology and function in vivo is challenging; therefore, imaging approaches, including magnetic resonance imaging (MRI)–volumetric analysis and positron emission tomography (PET) scans, are progressively recognized as valuable tools for the research of CP involvement in diseases such as MS.

Klistorner and colleagues have studied individuals with CIS, such as optic neuritis, individuals with RRMS, and healthy control (HC) participants and reported a significant transient increase in choroid plexus volume (CPV) in both CIS and RRMS groups during attacks of acute inflammation. However, the plexus volume returned to the pre-inflammatory status a few months later. Furthermore, individuals with CIS who converted to clinically definite MS (CDMS) during a ten-year follow-up period had significantly higher CPV compared with the HC group [21]. 

Ricigliano and colleagues, in a study including both 3.0-T brain MRI and PET results, reported a 32% CPV increase and higher mitochondrial translocator protein (TSPO) expression in the plexus of asymptomatic individuals with radiologically isolated syndrome (RIS) compared with HCs, further supporting the argument that plexus enlargement and BCSFB dysfunction may be an early phenomenon in the disease course, far beyond the MS diagnosis [17,22].

The same research group studied CPV changes in RRMS and progressive MS individuals and found that the plexus was more enlarged in individuals with gadolinium-enhancing lesions and correlated positively with the volume of T2-hyperintense white matter (WM) lesions in the RRMS group but not in progressive MS. Further, a higher ^18^F-DPA–714 binding protein uptake (which is a PET tracer sensitive to inflammation) was found in the normal-appearing white matter (NAWM) and thalami in the same group of individuals [15].

In the same line, CP enlargement was found in individuals with RRMS but not in individuals with neuromyelitis optica spectrum disorder (NMOSD) or HCs, suggesting that CP/ TIV (choroid plexus/total intracranial volume) ratio increases may be specifically related to the inflammatory processes in MS, which includes the infiltration of peripheral activated T lymphocytes versus the antibody-mediated anti-AQP4 inflammation in NMOSD disorder [16,19]. To our knowledge, no other studies have compared the differences in CP enlargement between MS and other neuroinflammatory or neurodegenerative disorders.

CP enlargement could overall be attributed to several different mechanisms. In experimental autoimmune encephalomyelitis (EAE), an animal model of MS, CPV increases were specifically related to microglial activation and infiltration, astrocyte activation (linked to WM lesions), increased albumin content in the CSF, and increased expression of genes related to T cell adhesion, differentiation, and activation [23,24]. This suggests that plexus enlargement could be due to increased infiltration by both CNS resident cells (microglia) and peripheral immune cells (T cells and activated macrophages), as well as increased BCSFB permeability and cellular edema within the plexus [11,23,25,26]. It is important to note that MRI-derived data (or PET imaging) on the microstructure of CP do not explain the functional mechanism underlying such morphological CP alterations. Whether plexus alterations are an early triggering event, an escalator of CNS inflammation, or a consequence of lesion expansion remains to be answered. Other hypotheses regarding CP enlargement in individuals with MS could involve endothelial immune proliferation, ependymal cell proliferation, CSF hypersecretion, oxidative stress, and hypoxia within the plexus, although the impact of those factors on plexus volume remains unknown [23,27,28,29]. Potential mechanisms of CP enlargement during active inflammation in MS are depicted in Figure 2. 

Taken together, the significant relationship arising between CPV and the degree of apparent (acute) inflammation in brain tissue (for example, gadolinium-enhancing lesions, ^18^F-DPA uptake in NAWM, thalami) opens the perspective to use plexus volume analysis as a promising MRI biomarker for MS disease activity, especially in the early MS stages, with a strong inflammatory component [15,16,23,30]. However, the temporal evolution of CPV over the course of the disease and its association with chronic inflammation, along with MS-related brain atrophy and neurodegeneration remain relatively poorly understood.

## 4. CP Enlargement as a Biomarker of Neurodegeneration

Klistorner et al. investigated the longitudinal changes in the plexus volume among RRMS individuals in an 8-year observational study and demonstrated that CPV increases gradually over the course of the disease. Specifically, the annual rate of CPV increase was 1.4% ± 1.2%/year [31]. Jankowska and colleagues also reported a subtle but constant increase in plexus volume during at the one-year follow-up in RRMS individuals [30]. Other studies found that CPV remained stable during the follow-up period [19,20,32]. That may suggest that CPs enlarge differently according to the phase of the disease and its inflammatory activity. 

According to Klistorner and co-workers, normalized CPV was robustly associated with deep gray matter (GM) volume loss (as measured by ventricular enlargement) and only slightly with cortical or whole brain atrophy. Furthermore, the progressive plexus enlargement was associated with the volume of chronic lesion expansion, but not with the number or volume of new lesion load [31,33]. These associations imply that the CP could be involved in (the beginning or sustaining) the low-grade inflammatory process at the edges of chronic periventricular MS lesions [20,33,34,35]. The expansion of chronic MS lesions has been implied as one of the main forces that drive MS progression [36]. Therefore, CP enlargement could contribute to the ongoing neurodegenerative component in long-standing MS [26,33,35]. Potential mechanisms of CP involvement in the neurodegenerative processes of MS are depicted in Figure 3. 

In another recent study, the enlarged and inflamed CP was significantly related to both cortical atrophy and deep GM atrophy in MS individuals but not in those with NMOSD (or HC participants), supporting the idea that common plexus and CSF immune-mediated pathological factors could be responsible for the accumulation of damage in tissues in direct contact with the CSF (such as periventricular and cortical regions) [19,37]. 

Raghib and co-workers recently demonstrated a positive correlation of higher plexus volume with retinal atrophy, using optical coherence tomography (OCT), and low retinal nerve fiber layer (RNFL) thickness has already been linked with diffuse abnormalities in NAWM and thalamic atrophy in RRMS [5,38]. However, it remains unclear how the enlarged CP could be implicated in the pathophysiology of retinal volume loss.

Other studies have not seen such definite correlations of CPV increase with brain atrophy. Akaishi and colleagues, in a cohort of RRMS and SPMS individuals, reported that normalized CPV was associated with the number and volume of WM lesions and WM atrophy but not total GM volume loss [39]. 

Brain atrophy in MS is the net effect of all destructive processes, but the main contributor is GM atrophy and to a lesser extent WM atrophy. Since GM atrophy reflects neurodegeneration and permanent disability, it is safer to conclude that CP enlargement reflects neuroinflammation, especially in RRMS individuals with active inflammatory profiles, but not neurodegeneration. 

## 5. Clinical Implications of Plexus Enlargement

So far, relatively little is known about the clinical relevance of CP inflammation. Klistorner et al. failed to show a correlation between CPV and Expanded Disability Status Scale (EDSS) changes [20]. Similarly, Ricigliano and colleagues did not find an association between the enlarged and inflamed CP and the severity of clinical disability, as measured by the EDSS score, but the plexus volume was markedly associated with the annual relapse rate in the RRMS group [15]. Jankowska et al. also confirmed that CP enlargement was connected to relapses in RRMS individuals, but it should be mentioned that it was a relatively small group (14 people) [30].

Bergsland and colleagues, in a study combining CP volumetric analysis and qualitative plexus pseudo-T2 mapping to assess CP inflammation, proposed that CP inflammation, particularly assessed by elevated pT2, was highly associated with future EDSS changes and disability progression, yet the association lost significance in volumetric plexus analysis, in that, normalized CP volume was not associated with disability worsening. Also, neither the CP enlargement nor the CP pseudo-T2 mapping showed significant correlations with the annual relapse rate [32]. 

Fleischer and colleagues, on the other hand, demonstrated a significant correlation between CP enlargement and clinical disability (EDSS score) in a cohort of CIS and RRMS individuals, both cross-sectionally and longitudinally, after 4-year follow-ups, supporting the clinical relevance of plexus volumetric analysis, among others [5,23,30]. Cognitive impairment, measured with the Symbol Digit Modalities Test (SDMT) score, was also associated with plexus enlargement [23]. 

Although the reason for this debate among researchers is currently unknown, it should be mentioned that the disease duration between the cohorts was remarkably different, and this may in part be the reason. In the study of Fleischer and colleagues, the mean disease duration was 3.1 years, while in Bergsland et al. the mean duration was 15.2 years. It may be the case that in long-lasting MS, the plexus fails to predict (upcoming or future) disability outcomes [32]. Other reasons may be the poor correlations generally reported between EDSS scores (especially <5) and inflammatory disease markers or the relatively small changes in EDSS scores in short-term follow-ups [40]. 

Furthermore, Fleischer and colleagues showed that in treatment naïve and dimethyl fumarate (DMF)-treated individuals with MS CPV increased over time, whereas in natalizumab-treated individuals CPV remained stable over time [23]. Natalizumab blocks the α4-β1 integrin on monocyte leukocytes and decreases the interaction with vascular cell adhesion molecule 1 (VCAM-1), preventing the migration of leukocytes through the disrupted BBB into the CNS [41]. In individuals with SPMS or inactive-PPMS, natalizumab is no longer effective, suggesting that leukocyte trafficking through the BBB no longer occurs through the CP. Therefore, CPV dynamics may serve as an MRI marker of the transition of RRMS to SPMS [42]. 

Further studies would be of outstanding interest to decide whether the volumetric status of CP and the cessation of CPV increase could be an indication of a transition to the progressive phase of the disease (SPMS), a biological event for which there is currently no biomarker. Nevertheless, studies recruiting higher numbers of RRMS and SPMS individuals would be necessary to determine the role of CPs in MS progression and transition to SPMS. Besides Natalizumab and DMF, there are no available data for the effect of other, currently approved, disease-modifying therapies (DMTs) for MS on CP volume; future studies might be necessary to address this issue. Table 1. summarizes the currently available studies on CP volumetric analysis, the soft wares used, as well as the main results of each study, respectively (Table 1). 

## 6. CP Volumetric Analysis, Limitations of Current Methodologies

The optimal neuroimaging sequences in order to visualize CP are the T1-weighted (T1W) MRI sequences enhanced with contrast medium (gadolinium). However, as a common rule, these agents are not routinely used in research settings; their use is limited to clinical settings when the benefits outweigh their risks. On the other hand, 3 Tesla T1W images are commonly acquired in research studies; however, the plexus’ intensity in voxels, which is similar to the neighboring GM structures, may confound the results and interpretation [44].

T1W sequences are commonly used to measure the CP within the lateral ventricles, the largest CP among brain ventricles. However, the vast majority of researchers do not include the infratentorial proportion, which corresponds approximately to one-third of total CPV; most probably because T1W images do not have enough resolution for the segmentation of the CPs of the third and fourth ventricle [5,19,32,45]. Higher resolution images or 7 Tesla MRI sequences may facilitate the volumetric CP analysis. 

For more accurate measurements, CP within lateral ventricles can be manually segmented, which is considered to be the gold standard technique, especially when applied to structural T1W MRI data [17,46,47,48]. However, this process is time-consuming and laborious in nature, and therefore, it is highly subject to human error. Moreover, it precludes large-scale volumetric analysis. The aforementioned reasons highlight the necessity to use accurate automatic CP segmentation techniques instead. Freesurfer software [49,50] has been used in the majority of previous studies for automatic CP segmentation, but future studies will evaluate its accuracy and consistency in assessments [15,32,39,44,46,48,51,52,53].

The absence of cut-off scores to evaluate the integrity of CP volumetric analysis raises the need to correct (or normalize) the CP value for the head size per individual [19,39]. Freesurfer software and Total Intracranial Volume (CP/TIV ratio × 1000) [15,20,39] have commonly been used for this correction [18]. Nevertheless, the development of new algorithms that will be solely used to segment CP would be beneficial in studying the structure and function of CPs in large-scale neuroimaging studies [44,53,54,55,56,57]. 

Of note, the presence of calcifications in CPs may affect the accuracy of segmentation [58]. Although CP calcifications are considered benign, without clinical implications, future studies are necessary in order to establish whether CP calcifications can lead to CP dysfunction and if their presence impacts the quality of volumetric analysis. Additionally, transient biological factors, such as dehydration or decreased protein levels, could also complicate volumetric analysis [59]. 

The CP contains estrogen, progesterone, and androgen receptors [60,61]. Akaishi and colleagues have reported that male individuals with MS may present higher plexus enlargement, while in other studies, the effect of sex on CPV was not significant [15,17,39,51]. Margoni and colleagues, on pediatric populations with MS, reported a higher CPV in females [60]. Further analysis may be needed to clarify the differences in CP volumetric analysis in males and females, respectively.

To sum up, multiple technical and biological factors interfere with CP volumetric analysis, which should be taken into consideration when evaluating its role in an already multifactorial disease like MS. 

## 7. Discussion

There has been a recent increase of interest in the role of CP in MS pathology due to its active role in regulating brain homeostasis, particularly the neuroinflammatory response of the CNS, and neural repair [6]. Initial research suggested that CPs are enlarged and inflamed on imaging modalities, particularly in highly active RRMS individuals [15]. Pharmacologic manipulation of the plexus with natalizumab prevented further enlargement, opening the perspective that CP dynamics may be a useful MRI marker of disease activity in early MS, while it may not be helpful in long-standing MS (SPMS) [23,39]. 

Although some studies provided evidence favoring an association between CP/TIV ratio and neurodegenerative changes, such as cortical thinning or deep GM atrophy in MS, other studies suggested that CP inflammation may not be the primary cause of neurodegeneration [19,33,39].

It is important to note that plexus alterations do not seem to be specific to MS. CP enlargement has also been reported in disease states linked with inflammation such as meningitis, depression, psychosis, schizophrenia, complex regional pain syndrome, epilepsy, ischemic stroke, hypoxia, obesity, as well as in primary neurodegenerative disorders, such as Alzheimer’s, frontotemporal dementia, Huntington’s disease, and Parkinson disease, as well as normal aging [7,46,47,48,51,56,62,63,64,65,66,67,68,69,70,71,72,73,74]. As such, CPV increase does not fulfill the established criteria of either a diagnostic or prognostic marker in MS [75], and up to now, there is no indication that it could replace the current clinical and radiological assessment of disease activity in any form of MS. 

In normal aging, the plexus shows a decline in all aspects of function. Structurally, the epithelium becomes flattened and atrophic, the stroma becomes rigid due to calcifications and fibrosis, and capillary walls become thicker due to lipofuscin granules [6,12,73]. Functionally, CSF production and clearance are also decreased with age [73]. Therefore, an increase in the CPV may be a compensatory or neuroprotective mechanism in response to the diminished ability to produce CSF [6,73]. Additionally, recent reports have demonstrated that enlarged ventricles and increased inflammation due to ischemic stroke or acute brain injury and subarachnoid hemorrhage (SAH) enhance the secretion of neuroprotective growth factors, resulting in volume increase of CP epithelium [73]. 

It has always been challenging to distinguish the neuroinflammatory and neurodegenerative components of MS, as the neuroinflammatory component can be both focal and diffuse, even in early MS, and it is associated with substantial axonal damage and neurodegeneration [76]. It has been suggested, however, that the pathophysiology of MS changes with disease progression; from primarily inflammatory to less inflammatory and predominantly neurodegenerative in the late stages of the disease [77]. In progressive MS, additional mechanisms of axonal loss coexist, such as ionic overload, mitochondrial dysfunction, iron dysregulation, and glutamate excitotoxicity, which are at least partly independent of the mild apparent inflammation within the CNS [78,79]. In addition to that, and due to the multifunctional role of CP in brain homeostasis, it is difficult to distinguish the dominant purpose of volume enlargement, even in normal aging [73]. 

In post-mortem studies, the CP stroma of RRMS individuals has shown apparent inflammatory modifications, with infiltrating T cells, activated macrophages/microglia, activated complement deposition, severe disruption of tight junctions in the choroidal epithelial cells, and overexpression of adhesion molecules, such as VCAM-1. In individuals with progressive MS (either primary or secondary), this immune infiltration was mild and similar to the control group, only including granulocytes, NK cells, and clusters of T cells (CD8+ T cells) [11,29]. Accordingly, abnormalities found in relapsing MS CSF included increased numbers of T cells, higher CD4+/CD8+ ratio, memory T cells, memory B cells, antibody-secreting plasma cells, and neutrophils [11]. Contrary to that, the CSF in progressive MS, included only B cells, and a few granulocytes [29]. Rodríguez-Lorenzo and co-workers suggested that the plexus may be only marginally involved in immune cell trafficking into the CNS in the neurodegenerative-dominant phase of the disease [29]. Instead, the presence of B-cells in the meninges, seen in individuals with progressive MS, may be the source of these cells in the CSF [4,13,80]. Dysfunction of the hypoxia-induced factor 1 signaling pathway, reflecting altered secretory and neuroprotective properties of the plexus, has also been described in individuals with progressive MS [29]. However, to what extent these hypoxic, inflammatory, and secretory changes may influence CPV assessment in progressive MS remains unknown [15].

All things considered, it may be suggested that the morphology and behavior of CP change with disease progression from primarily inflammatory to less inflammatory and predominantly hypoxic in progressive MS stages [15,29]. In MS lesions (active or inactive plaques), mitochondrial dysfunction in neurons and oligodendrocytes (OLDs) results in the production of free oxygen radicals (ROS), which may either remain restricted to the brain parenchyma or be diffused into the CSF, reaching the CP [4]. Perhaps the focus and the amount of knowledge gathered on the highly inflammatory RRMS phase creates a predisposition towards the “outside–in” model and distracts us from studying the underlying neurodegeneration occurring in progressive phases, which is more consistent with the “inside–out” model for MS [81].

## 8. Conclusions

Through this review, we aimed to improve our understanding regarding the role of CP in the context of both inflammatory and progressive MS. Overall, we suggest that the assessment of CPV should mainly be focused on the early disease stages, such as RIS, CIS, and RRMS, and it should be taken into consideration with caution in long-standing MS stages. In that respect, further longitudinal research with more frequent serial imaging in individuals in their early disease stage may be useful to clarify the role of the enlarged CP in MS. Moreover, higher CPVs could potentially predict which individuals with RIS or CIS are at higher risk for increased disease activity. Last but not least, this line of research could provide unconventional pharmacological targets for future interventions, which might either prevent BCSFB disruption or even repair BCSFB dysfunction during the early stages of the MS spectrum.

## Figures and Tables

**Figure 1 healthcare-12-00768-f001:**
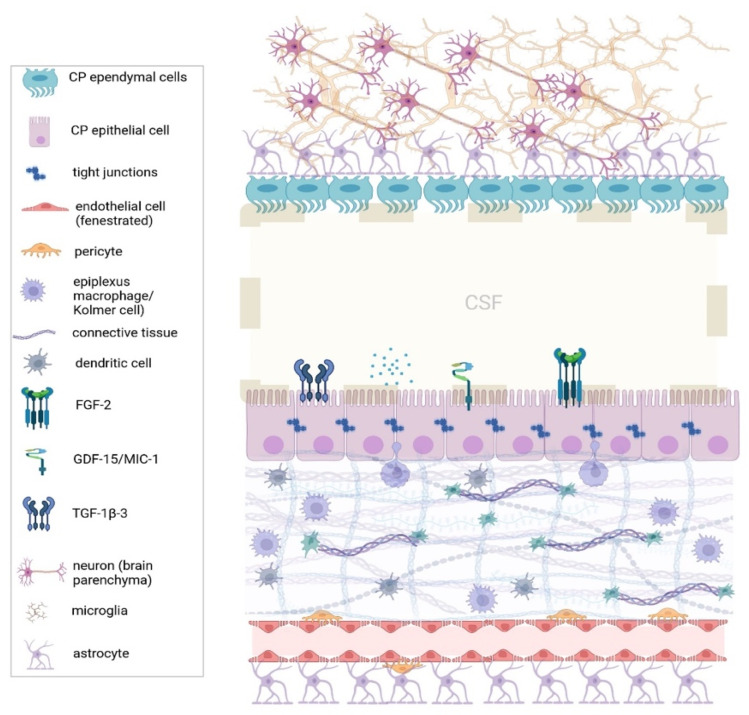
The architecture of the choroid plexus. Normally, the choroid plexus is an active regulator of CNS homeostasis by producing CSF and regulating the immune cell trafficking into the brain parenchyma. Immune cells reside in the plexus stroma, mostly epiplexus macrophages (or Kolmer cells), which project between the epithelial cells, and dendritic cells. The blood–CSF barrier (or plexus epithelium) tightly regulates the passage of molecules and solutes from the systemic circulation into the CNS. (CNS: Central Nervous System, CSF: Cerebrospinal fluid, FGF-2: Fibroblast growth factor-2, GDF-15: Growth/differentiation factor 15, MIC-1: macrophage inhibitory cytokine 1, TGT-1β-3: Tissue Growth Factor-1β-3.) Created with BioRender.com.

**Figure 2 healthcare-12-00768-f002:**
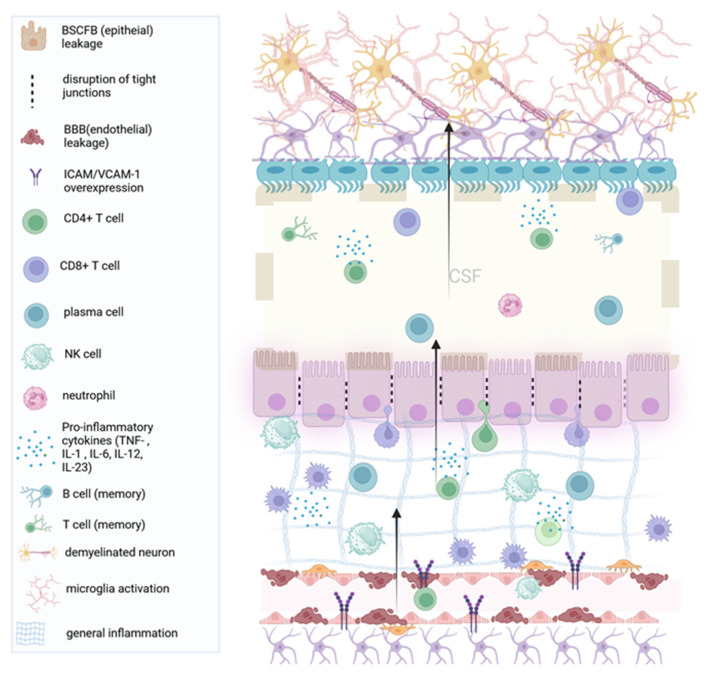
Acute inflammation and choroid plexus edema (enlargement): In the inflammatory phases of the disease, general inflammation leads to demyelination. The choroid plexus endothelium recruits pro-inflammatory CD4+ T cells through the upregulation of adhesion molecules such as VCAM1. The accumulation of the plexus stroma with T cells leads to swelling of the plexus (local edema), while immune cells cross the plexus epithelium (and the permissive ependyma) into the brain parenchyma. (BCSFB: Blood cerebrospinal fluid barrier, BBB: Blood–brain barrier, IL: Interleukin. VCAM-1: Vascular cell adhesion molecule 1). Created with BioRender.com.

**Figure 3 healthcare-12-00768-f003:**
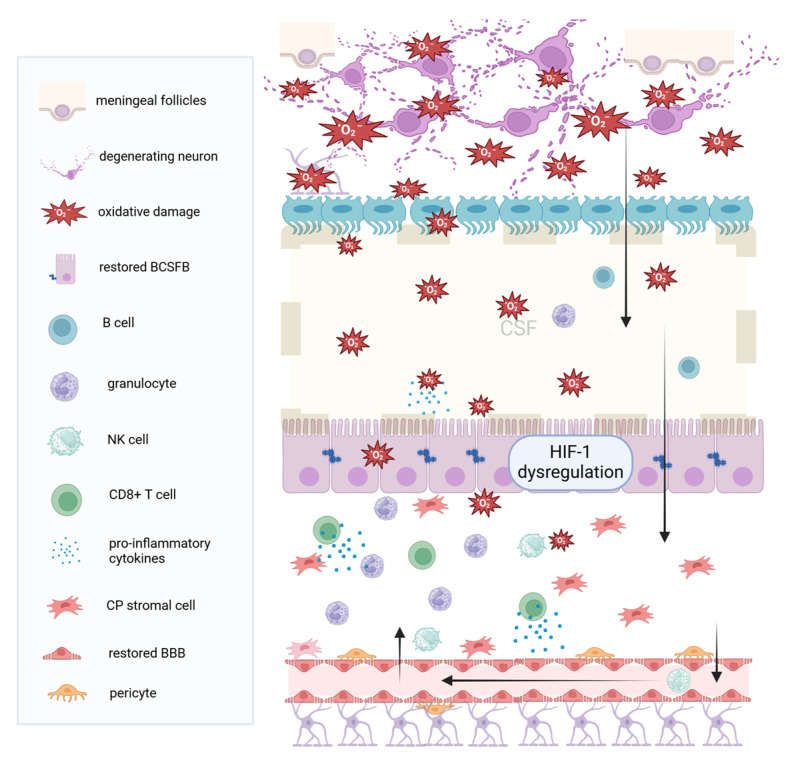
Chronic inflammation and choroid plexus hypoxia. In progressive MS (either secondary or primary progressive), inflammation is mild, and neurodegeneration prevails. Damage signals from the brain, such as ROS and iNOS, leak through the ependymal cells, into the ventricular CSF, towards the choroid plexus. These signals lead to oxidative stress, activation of the HIF 1 signaling pathway, and virtual hypoxia at the plexus. The BCSFB and the BBB are intact or restored. In response to chronic intrathecal inflammation and ROS, the plexus produces cytokines, leading to the recruitment of granulocytes and neutrophils, which further contribute to inflammatory cytokines secretion. T cells, particularly the CD8 subset, also accumulate in the plexus stroma, together with NK cells through the blood–brain barrier, into the CSF, towards the periventricular brain tissue. (MS: Multiple Sclerosis, CSF: Cerebrospinal Fluid, BCSFB: Blood cerebrospinal fluid barrier, BBB: Blood–brain barrier, NK: Natural Killer, CP: Choroid Plexus, ROS: Reactive oxygen species iNOS: Inducible nitric oxide synthetase, HIF-1: hypoxia-induced factor 1). Created with BioRender.com.

**Table 1 healthcare-12-00768-t001:** Studies that measured choroid plexus volume in multiple sclerosis.

Reference	MS Sample	Segmentation	Volumetric Analysis Software	Outcomes
Müller et al., 2022 [16]	180 PwMS (RRMS, SPMS) 98 NMOSD 94 HCs and 47 migraine patients	Automatically	GitHub	Significantly enlarged CPs in PwMS but not NMOSD or HCs and migraine patients.
Ricigliano et al., 2022 [17]	27 RIS 97 CDMS 53 HCs	Manually	ITK-SNAP	Individuals with RIS had 32% larger CPs, similar to individuals with CDMS.
Ricigliano et al., 2021 [15]	61 RRMS 36 PMS 44 HCs	Manually& automatically	ITK-SNAP & FreeSurfer	CPs are enlarged (35% increase) and inflamed in PwMS, particularly in the RRMS group.
Klistorner et al., 2022 [31]	49 RRMS 40 HCs	Semi-automatically	JIM 9 (Xinapse Systems, Essex, UK)	CPs are enlarged in RRMS individuals versus HCs, and the baseline CP/TIV ratio predicts subsequent expansion of chronic periventricular MS lesions.
Jankowska et al., 2023 [30]	14 RRMS (Treg group) 22 RRMS Treatment-naïve 16 HCs	Automatically	BrainMagix	CPs are enlarged and inflamed in individuals with RRMS with a strong inflammatory component.
Preziosa et al., 2024 [43]	129 PwMS 73 HCs	Automatically	FSL-SIENAX	CP enlargement may contribute to the pathophysiology of cognitive impairment and fatigue in PwMS.
Raghib et al., 2024 [5]	95 RRMS 26 HCs	Automatically	FreeSurfer	tCPV was found to be strongly associated with clinical disability and retinal atrophy in RRMS individuals.
Akaishi et al., 2024 [39]	89 RRMS 13 SPMS 41 HCs	Automatically	FreeSurfer	CP enlargement in PwMS was associated with WM atrophy but not GM atrophy.
Bergsland et al., 2023 [32]	118 RRMS 56 PMS 56 HCs	Manually	ITK-SNAP	CP enlargement is clinically relevant and may have a role in driving disability worsening (5-year follow-up).
Chen et al., 2023 [19]	51 PwMS 42 NMOSD 56 HCs	Automatically	FreeSurfer	Enlarged CPs were found to be related to cortical atrophy and subcortical deep GM atrophy in PwMS, but not in individuals with NMOSD.
Wang et al., 2023 [35]	99 RRMS 60 HCs	Manually	ITK-SNAP	The enlargement of CP in RRMS correlated positively with deep GM atrophy, but not with cortical atrophy. Also, the bigger plexus volume was associated with higher disability (EDSS) and lower cognitive scores (SDMT).
Klistorner et al., 2023 [21]	44 CIS 50 RRMS 50 HCs	Semi-automatically	JIM 9 (Xinapse Systems, Essex, UK)	Transient CP enlargement during attacks of acute inflammation, in early MS.
Fleischer et al., 2021 [23]	330 PwMS 57 HCs	Automatically	Freesurfer	CP enlargement is robustly associated with disability progression. Additionally, pharmacological manipulation of BCSFB with Natalizumab prevented further enlargement.

PwMS: patients with multiple sclerosis, CP: choroid plexus, RRMS: relapsing–remitting multiple sclerosis, SPMS: secondary progressive multiple sclerosis, PMS: progressive multiple sclerosis, NMOSD: neuromyelitis optica spectrum disorders, HCs: healthy controls, RIS: radiologically isolated syndrome, CDMS: clinically definite multiple sclerosis, CIS: clinically isolated syndrome, TIV: total intracranial volume, tCPV: total choroid plexus volume, EDSS: expanded disability status scale, SDMT: symbol digit modalities test, BCSFB: blood–cerebrospinal fluid barrier, WM: white matter, GM: gray matter.

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
