# Peer review of "The Time Trajectory of Choroid Plexus Enlargement in Multiple Sclerosis"

_healthcare, 2024, doi:10.3390/healthcare12070768_

Round 1
Reviewer 1 Report
Comments and Suggestions for Authors
This manuscript provides a comprehensive discussion about the significance of choroid plexus enlargement in multiple sclerosis. The paper is well-organized and summarizes most of the recent findings on choroid plexus enlargement in different types of MS. Addressing the concerns and questions below will help clarify and better understand the study:
1) Choroid plexus enlargement is not specific to multiple sclerosis but is also found in other neurodegenerative disorders. Is there any evidence showing the difference in enlargement between MS and other neurodegenerative disorders? Does it also relate to neuroinflammation in Alzheimer’s disease or other neurodegenerative disease?
2) In terms of CSF composition and characteristics, are they different among different types of MS?
3) Are there any therapeutic methods or medicines that can prevent choroid plexus enlargement?
4) What are the differences in the receptors on the choroid plexus between MS patients and healthy controls?
5) Besides neurodegenerative diseases, are there other gene mutation diseases that also show choroid plexus enlargement?
Reviewer 2 Report
Comments and Suggestions for Authors
In this review the authors attempt to clarify the pathological significance of the temporal CP volume (CPV) changes in MS and highlight the pitfalls and limitations of CP volumetric analysis. The topic is clinically interesting, but several issues should be raised.
Please describe the methodology in more detail. How many researchers were involved in the literature review, how the publications for this review were selected? I suggest to show screening protocol as flowchart.
Furthermore, the review clearly shows that CPV does not meet the criteria for an MS biomarker and there is no indication that it will replace the current radiological assessment of disease activity, which should be clearly mentioned in the manuscript. In fact, CPV is another non-MS specific parameter that may only be supportive in selected clinical circumstances.
